# A Synergetic Strategy for Brand Characterization of Colla Corii Asini (Ejiao) by LIBS and NIR Combined with Partial Least Squares Discriminant Analysis

**DOI:** 10.3390/molecules28041778

**Published:** 2023-02-13

**Authors:** Ziyi Xia, Xiaoqing Che, Lei Ye, Na Zhao, Dongxiao Guo, Yanfang Peng, Yongqiang Lin, Xiaona Liu

**Affiliations:** 1College of Integrated Traditional Chinese and Western Medicine, Binzhou Medical University, Yantai 264003, China; 2Shandong Runzhong Pharmaceutical Co., Ltd., Yantai 256603, China; 3Key Laboratory of Xinjiang Phytomedicine Resources and Utilization in Ministry of Education, School of Pharmacy, Shihezi University, Shihezi 832002, China; 4Shandong Institute of Food and Drug Inspection, Jinan 250101, China; 5Pharmacy Faculty, Hubei University of Chinese Medicine, Wuhan 430065, China

**Keywords:** laser-induced breakdown spectroscopy, near-infrared spectroscopy, data fusion, *Colla Corii Asini*, brand characterization

## Abstract

A synergetic strategy was proposed to address the critical issue in the brand characterization of *Colla corii asini* (Ejiao, CCA), a precious traditional Chinese medicine (TCM). In all brands of CCA, Dong’e Ejiao (DEEJ) is an intangible cultural heritage resource. Seventy-eight CCA samples (including forty DEEJ samples and thirty-eight samples from other different manufacturers) were detected by laser-induced breakdown spectroscopy (LIBS) and near-infrared spectroscopy (NIR). Partial least squares discriminant analysis (PLS-DA) models were built first considering individual techniques separately, and then fusing LIBS and NIR data at low-level. The statistical parameters including classification accuracy, sensitivity, and specificity were calculated to evaluate the PLS-DA model performance. The results demonstrated that two individual techniques show good classification performance, especially the NIR. The PLS-DA model with single NIR spectra pretreated by the multiplicative scatter correction (MSC) method was preferred as excellent discrimination. Though individual spectroscopic data obtained good classification performance. A data fusion strategy was also attempted to merge atomic and molecular information of CCA. Compared to a single data block, data fusion models with SNV and MSC pretreatment exhibited good predictive power with no misclassification. This study may provide a novel perspective to employ a comprehensive analytical approach to brand discrimination of CCA. The synergetic strategy based on LIBS together with NIR offers atomic and molecular information of CCA, which could be exemplary for future research on the rapid discrimination of TCM.

## 1. Introduction

Ejiao (*Colla Corii Asini*, CCA) made from donkey hide has been widely used as a precious traditional Chinese medicine (TCM) in China for more than 2000 years [1,2]. As recorded in ancient and classic TCM monographs, CCA shows great efficacy in enriching blood and staunching bleeding [3]. Nowadays, multiple brands of CCA have been manufactured by different pharmaceutical factories in China. Quality difference of CCA exists among different brands, affecting nutritional value and market price. The nutritional properties of CCA are affected by such factors, as geographical origins, raw materials, water sources as well as pharmaceutical processes. Among these factors, geographical origin mainly confers nutritional qualities to CCA. Traditionally, CCA is evaluated according to its producing areas [3]. For example, Dong’e Ejiao (DEEJ) produced in Dong’e county of Shandong province has been acknowledged as authentic and top-quality CCA products. DEEJ is well known as one of the intangible cultural heritage resources. However, a similar manufacturing process of CCA caused similar appearance characteristics and physical features. It is difficult to distinguish DEEJ from CCA of other brands. A critical issue in the quality control of Ejiao is brand or manufacturer identification. Therefore, a way of discriminating against CCA is needed for the purpose of brand protection and competitive advantage.

To date, multiple analytical techniques have been successfully prioritized for quality assessment and differentiation of CCA, such as vibrational spectroscopic techniques (Fourier transform infrared (FTIR) spectroscopy and near-infrared (NIR) spectroscopy [4,5], liquid chromatography mass-spectrometry [6,7], gas chromatography-mass spectrometry (GC-MS) [4], and Polymerase Chain Reaction (PCR) [8], etc. The truth is that data obtained from different analytical techniques can provide complementary chemical information thus improving the qualitative or quantitative analysis performance [9,10,11,12]. However, these approaches applied only one analytical tool.

Spectroscopic techniques are appealing tools for fast quality control of traditional Chinese medicine (TCM) with no laborious sample preparation. For example, laser-induced breakdown spectroscopy (LIBS) and near-infrared spectroscopy (NIR) have many advantages such as minimal sample pretreatment, fast detection, less destruction, environmentally friendly and cost-effective properties of the analytical procedure, have been uesd in pharmaceuticals, food, and agriculture filed [13,14,15]. LIBS is a spectrochemical sensor technology for the acquisition of simultaneous multi-elemental analysis based on atomic emission [16,17]. In LIBS, a pulsed laser beam is focused on the sample to vaporize the sample and induce the formation of a plasma. The emission of the plasma containing excited atoms and ions present in the sample is collected and analyzed through an optical system and a spectrometer in order to extract the spectroscopic information [18]. Qualitative and quantitative information can be obtained by identification of the spectral lines and analysis of their peak intensities, respectively. LIBS has the unique advantages of fast analysis, in-situ or remote measurements, and ability to analyze various samples such as solids, liquids, and gases with little or no sample preparation [17]. It is well accepted that some inorganic elements of TCM play significant roles in biological activity, which are responsible for the clinical effect [19]. Hence, elemental fingerprints are useful to identify the geographical origin and species of TCM.

Another promising technique is NIR, which has been successful in the quality control of food and drug. As a non-destructive technique, NIR employs measurements of photon energy to obtain qualitative and quantitative information based on the interaction of the sample with NIR radiation [20,21]. It covers the wavelength range of 800–2500 nm and mainly reflects vibration information of hydrogen bonds in a molecular structure, such as carbon–hydrogen (C-H), oxygen–hydrogen (O-H), nitrogen–hydrogen (N-H) [22,23]. Particularly, hydrogen bonds own high spectral stability in the NIR region, which aid in qualitative and quantitative analyses.

The combination of LIBS and NIR could provide complementary information on unknown species, involving not only the elemental compositions but also the molecular components. Nowadays, the two techniques have been prevailing analytical tools for quality control of TCM with the rapid development of powerful computers and data analysis tools [24,25,26]. In addition, the potential applications of LIBS coupled with NIR on analyses of food and vegetable achieve considerable success [18,27]. However, brand characterization of CCA by combining LIBS with NIR has not been reported yet.

Usually, the application of LIBS and NIR spectroscopy in classification analysis needs the assistance of chemometrics approaches to construct the discriminant model. Among these approaches, partial least squares discriminant analysis (PLS-DA) is an effective algorithm used to optimize separation between different groups of samples [28]. PLS-DA is based on the classical PLS regression algorithm, which models the response matrix Y (class membership) through the predictive matrix X (original data) [28,29]. The Y matrix in PLS-DA is a set of binary variables (0 and 1), representing the membership of each sample. The main advantage of PLS-DA is that it can offer a graphical visualization and understanding of the different data patterns and relations by latent variables (LVs) and loadings [30].

The main objective of the present study was to classify CCA samples according to their species. CCA samples mainly contain organic and inorganic components including collagen, proteins, amino acids, polysaccharides, trace elements, water, etc. [3]. We focused on the application of a synergetic strategy to build robust classification models for brand characterization of CCA collected from different pharmaceutical factories in China. LIBS and NIR were fully investigated to offer atomic and molecular information on CCA samples. In this work, PLS-DA was chosen for model development based on LIBS and NIR spectroscopy processed separately. Considering the potential comprehensive information of CCA species provided by LIBS and NIR, a low-level data fusion strategy was also employed. After spectral pretreatment, two types of PLS-DA models were established, namely, single models and data fusion models. Then, the classification performance of PLS-DA models was evaluated by three statistical indicators such as total accuracy, sensitivity (Se), and specificity (Sp) [31,32].

The methodology is found to be helpful for the quality control of CCA, and exhibits the promise of addressing similar needs in other precious TCM products in the future.

## 2. Results and Discussion

### 2.1. Spectral Features of LIBS and NIR

The normalized LIBS spectra of the CCA sample is shown in Figure 1. The elemental compositions of CCA were identified and marked according to the NIST (US, National Institute of Standards and Technology) database [33]. LIBS spectra are quite complex due to the emission of multi-elements from samples. The characteristic spectral lines of Ca, Na, Mg, K, and Fe were observed in the CCA LIBS spectra. In addition, spectral lines of organic compositions, such as C, N and O can also be observed, which are actually both from CCA samples and the air. Additionally, some molecular spectra, such as C-N emissions appear in the LIBS spectrum. While the C-N bands arise from the interaction of carbon with atmospheric nitrogen. Selecting the appropriate characteristic spectra is significant to the classification results. The 47 spectral lines including 13 elements and four C-N molecular bands are summarized in Table 1 and chosen for further analysis.

Figure 2 shows plots of the original NIR spectra of CCA samples. The major features of the raw NIR spectra exhibit no distinct differences among different samples except the transmittance (indirect reflecting the absorption intensity). The NIR spectra reflects the organic molecules’ information containing molecular bonds, such as C-H, O-H, N-H, etc [34]. Three characteristic absorption bands can be viewed at 5784 cm^−1^, 6650 cm^−1^, 8417 cm^−1^, which are abundant in the molecular structures of amino acids, peptides, proteins, etc. In addition, the relatively slow absorption peak appeared at approximately 5151 cm^−1^, which may correspond to the frequency of the O–H bond in water molecules. The NIR spectroscopy-based fingerprint contains complex attributes of the CCA samples sufficiently, which are related to brand characterization.

### 2.2. PLS-DA Model from LIBS Data

The PLS-DA model was built on a single LIBS data block using the selected variables (listed in Table 2). When the number of LVs set as 11, the classification accuracy of the PLS-DA model reached the max value. Hence, the optimal number of LVs was selected as 11. Subsequently, the PLS-DA model was constructed to validate the training set and predict the testing set. Figure 3 displays the classification result of the PLS-DA model using the LIBS spectra. As can be seen from Figure 3, all samples are properly classified in the training set while the DEEJ samples are misclassified as non-DEEJ samples and one non-DEEJ sample is misclassified as a DEEJ sample in the testing set. The total accuracy values of the PLS-DA model are 100% and 85.18% for different class samples in the training set and the testing set, respectively. The sensitivity values of the PLS-DA model are 100% and 91.67% for different class samples in the training and testing sets, while the specificity values are 100% and 92.30% for different class samples, respectively. The misclassified sample in the top right corner of Figure 3 was sample No. 74 (belonging to Shandong Fupai Ejiao Co., Jinan, China, Ltd., listed in Table 2). We compared the LIBS spectra. It was found that the LIBS spectra of non-DEEJ sample No. 74 was very similar to DEEJ sample No. 38 (listed in Table 2), which indicated that they have the same elemental compositions and similar peaks intensity.

### 2.3. PLS-DA Model from NIR Data

The performance of PLS-DA models for NIR spectra with different pretreatment methods is shown in Table 3. The total accuracy values of the PLS-DA model based on raw NIR spectroscopy are 100% and 88.89% for the training set and the testing set, respectively. The sensitivity values of the PLS-DA model are 100% and 82.35% for different class samples, while the specificity values are 100% both in the training set and testing set, respectively. As shown in Figure 4, all samples are properly assigned in the training set but four Non-DEEJ samples are incorrectly assigned as DEEJ samples in the testing set, which means that, except for these four incorrectly classified samples, the model correctly predicted the class of the other samples.

Screening of specific NIR wavebands was employed to increase the classification performance between DEEJ and Non-DEEJ samples. The screened wavebands include 5500–9000 cm^−1^, which might represent the special spectroscopic characteristic of the discriminatory markers. The discrimination ability of PLS-DA models building with original and different pretreatments of NIR spectra were computed and compared. As shown in Table 3, all PLS-DA models constructed with NIR spectra show good prediction performance. Meanwhile, the discrimination potential in the testing set is better than that in the testing set. The PLS-DA model established by the raw NIR spectra exhibits satisfactory performance. The total accuracy values are 100% and 96.3% for different class samples in the training set and in the testing set, respectively. The sensitivity values are 100% and 93.33% for different class samples in the training set and testing set, respectively, while specificity values reach 100% for different class samples in the training set and in the testing set, respectively. Theoretically, as might be expected, PLS-DA models built on the NIR data by different data preprocessing methods can improve the discrimination. But the models obtained by Savitzky-Golay smoothing (SG) and first derivative (1st d) pretreatment do not show better performance than that of screened original NIR spectra.

Among all the discriminant models, the PLS-DA model with NIR spectra pretreated by the multiplicative scatter correction (MSC) method offers perfect performance than the others. When the optimal number of LVs set as 10, this PLS-DA model provides correct classification for all samples both in the training and testing sets. The PLS-DA discriminatory information of NIR spectra pretreated by the MSC method is shown in Figure 5. Therefore, the MSC method is identified as the optimal pretreatment method for NIR spectra. The capability of NIR spectroscopy for CCA authentication can attribute to the differences in the amount of hydrogen bonds (C-H, O-H, and N-H) absorption for the organic compounds presenting in CCA samples.

### 2.4. PLS-DA Model from Data Fuion of LIBS and NIR

Data fusion is the integration of data blocks from different analytical technologies into a single model [35,36]. Though, single LIBS or single NIR can provide good discrimination ability. The low-level fusion strategy based on concatenating LIBS and NIR spectra was also investigated in this work. The discrimination ability of PLS-DA models based on the low-level fusion with different pretreatments is shown in Table 4. The total accuracy values of PLS-DA models based on LIBS combined with the screened original NIR spectra are 98.04% and 96.30% for different class samples in the training set and testing set, respectively. The sensitivity values are 96.15% and 93.33%, whereas specificity values are 96.15% and 100% for different class samples in the training sets and in the testing sets, respectively.

Taking the classification efficiency into account, SNV and MSC were the best data preprocessing methods for discriminating DEEJ from Non-DEEJ, which can eliminate redundant information and manifest differences among samples. When the optimal number of LVs set as 10, the total accuracy values of the PLS-DA model based on low-level data fusion with SNV data pretreatment reach 100% for different class samples both in the training sets and testing sets. The sensitivity values and specificity values of the PLS-DA model are 100% for different class samples both in the training sets and testing sets. The data fusion model also exhibits excellent discrimination performance with MSC data pretreatment. When the optimal number of LVs set as 11, the total accuracy values, sensitivity values, and specificity values all reach 100%, indicating no sample is misclassified.

## 3. Materials and Methods

### 3.1. Sample Preparation

Seventy-eight CCA samples were collected from different manufacturers in China, including forty DEEJ samples and thirty-eight CCA samples from other factories called Non-DEEJ. Sample storage may influence the CCA quality. As a drug, CCA was sealed with a box. Thus, all samples were stored in a cool, dark area before analysis. The detailed information on all CCA samples is listed in Table 1. Due to the solid and glossy surface of CCA samples, no special sample preparation was performed in LIBS and NIR experiments.

### 3.2. Sample Measurement

The LIBS system used for CCA analysis comprises a Q-switched Nd: YAG laser operating at 1064 nm with a pulses duration of 3–5 ns at a frequency of 2 Hz. Echelle spectrometer (LTB, ARYELLE 150, Berlin, Germany) records the spectrum from 193 to 840 nm with an approximate resolution of 0.1 nm. The laser beam was focused on the sample surface by a microscope objective lens system with an 80 mm focal length (Avantes UV-74). The focused spot size on the target surface was about 100 μm in diameter. An optical fiber was connected to the spectrometer with an intensified gated CCD camera (LTB ARYELLE 200 and iStar from Andor Technology) to disperse and record the spectra. A laser pulse energy of 40 mJ, a gate width of 100 μs, and a delay time of 0.6 μs resulted in the maximum signal-to-noise ratio, then being selected for LIBS experiments. All LIBS experiments were performed in ambient air. “LTB-Sophi” software was used to record the emission spectrum of the species and extract analytical information. Four laser shots per location of twelve sampling locations were collected and spectra averaged aiming to limit the relative standard deviation of the peak-intensity ratios.

NIR spectra were collected by Antaris Nicolet FT-NIR system (Thermo Fisher Scientific Inc., Waltham, MA, USA). All spectra were measured in integrating sphere diffuse mode. Each spectrum was an average of 64 scans across the wavenumbers from 10,000 to 4000 cm^−1^ with a resolution of 4 cm^−1^ at ambient temperature. The spectra was recorded as Log(1/R) using air as a reference.

### 3.3. Spectral Pretreatment

The original LIBS spectra are often affected by various sources of interference from light scattering, instrument noise, and background [27]. Thus, conventional minimum-maximum normalization was employed to transform the original LIBS data to the range of 0–1 [27]. The raw spectra acquired from the NIR spectrometer are often characterized by baseline shift and unwanted spectral variation produced from light scattering effects, variation in optical path length, random noise, sample matrix, and others. In order to build a robust and stable model, data pretreatment must be taken prior to data analysis to minimize interference of NIR spectra. Here, four basic NIR spectral pretreatment methods were applied to reduce interference and optimize model performance, including standard normal variate (SNV), multiplicative scatter correction (MSC), Savitzky-Golay smoothing (SG), first derivative (1st d). Moreover, the optimal pretreatment method is selected by the highest classification accuracy rule in both training and testing sets.

### 3.4. Multivariate Analysis and Latent Variable Analysis

The spectral preprocessing and model calculation steps were performed by the Unscrambler 9.7 software (CAMO software AS, Oslo, Norway) and homemade routines in Matlab version R2009a with Statistical Toolbox. Partial least squares discriminant analysis (PLS-DA) algorithm was investigated for a single data block or fusion block respectively. To obtain robust multivariate calibration models, 78 samples were split into two sets of training (calibration) and testing (validation) sets in a ratio of 2:1 by KennardStone (KS) algorithm. Training set is used for building the model while the testing set is used for assessing the robustness of the model [29]. Before PLS-DA, several pretreatments were performed on LIBS and NIR spectra. It is generally known that the number of latent variables (LVs) is a critical parameter that is obtained from the evaluation of the class border by Bayesian decision [30]. The optimal number of LVs for each model was determined by 10-fold cross-validation based on the highest classification accuracy. Then the optimized model was used for the testing set. Besides total accuracy (Tc), sensitivity (Se) and specificity (Sp) were mainly computed to evaluate the performance of the established PLS-DA model based on preprocessed data. Sensitivity, also called the true positive rate (TPR), represents the positive correctly classified samples to the total number of positive samples as in Equation (2). Whereas specificity, also called the true negative rate(TNR), is expressed as the proportion of the correctly classified negative samples to the total number of negative samples as in Equation (3).

The three evaluation parameters were calculated as follows:Tc = (*TP* + *TN*)/(*TP* + *FP* + *FN* + *TN*)(1)

Se = *TP*/(*TP* + *FN*)(2)

Sp = *TN*/(*FP* + *TN*)(3)

where, *TP* and *TN* indicate the number of true positive samples and true negative samples, respectively. While *FP* and *FN* indicate the number of false positive samples and false negative samples. For each class, a positive sample means itself, and a negative sample is the other class. The sensitivity corresponds to the model’s ability to correctly predict the class of the samples, while specificity reflects its ability to incorrectly prediction of samples from other classes [29]. Perfect discrimination of the PLS-DA model yields the highest values of total accuracy, sensitivity, and specificity, indicating all samples are successfully classified.

### 3.5. Data Fusion Strategy

Data fusion can produce more accurate information by integrating multiple complementary data sources. Basically, it can be categorized into three levels: low level, mid level, and high level [37,38,39]. Low-level data fusion strategy was adopted in this work. The scheme of data analysis in this experiment is shown in Figure 6. To prepare for low-level fusion, raw data from all CCA are straightforwardly concatenated into a new data matrix. Before data concatenation, all variables were normalized to the same scale aimed at rectifying the dimensional imbalance between raw data matrices.

## 4. Conclusions

In this study, the feasibility of individual and combining information from LIBS and NIR techniques was researched aiming to address brand discriminant of Ejiao. LIBS and NIR spectroscopies bring atomic and molecular information of species. The discrimination results of PLS-DA models showed that individual techniques had good performance at authenticating brands of CCA. Especially, the PLS-DA model based on the single NIR data pretreated with the MSC method achieved a perfect classification effect, and it obtained 100% correct classification of each class for all samples. A data fusion strategy was also investigated for complementary information from two spectroscopic techniques. Data fusion models based on LIBS and NIR with SNV and MSC pretreatment also show good predictive ability. In general, the synergetic strategy of LIBS and NIR spectra demonstrated the possibility of brand discrimination of Ejiao, and it can be used to provide a reliable method for brand protection or geographical authenticity evaluation of TCM.

## Figures and Tables

**Figure 1 molecules-28-01778-f001:**
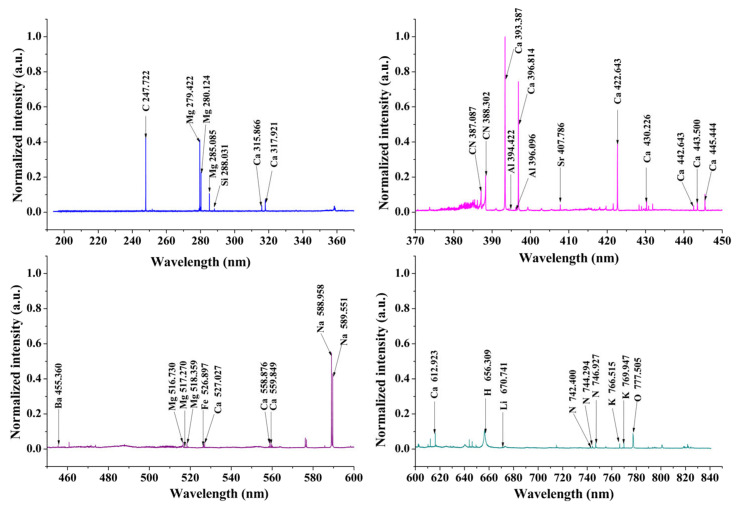
The representative normalized LIBS spectrum of the CCA sample.

**Figure 2 molecules-28-01778-f002:**
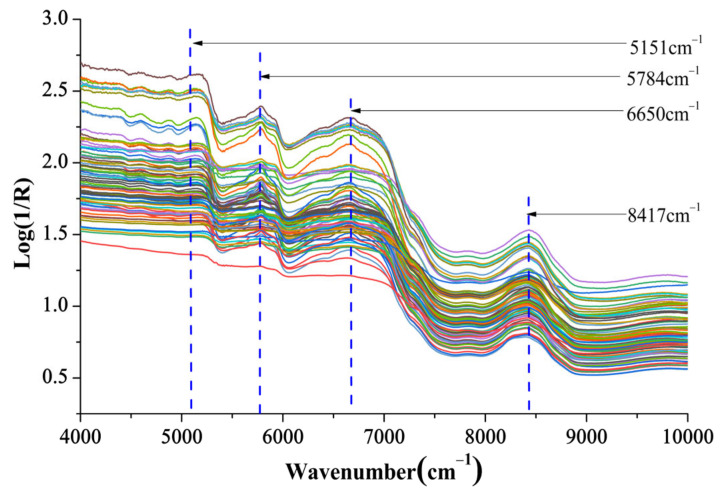
NIR spectra of Ejiao samples.

**Figure 3 molecules-28-01778-f003:**
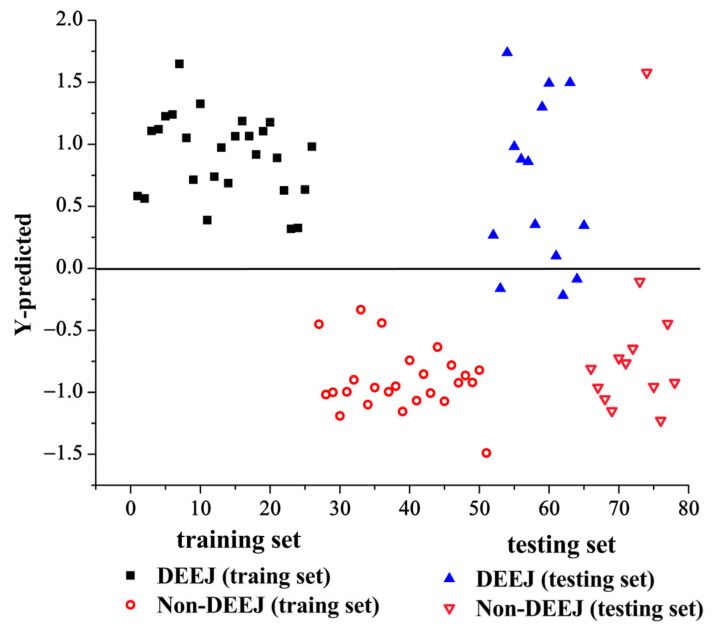
The PLS-DA model using the LIBS spectra.

**Figure 4 molecules-28-01778-f004:**
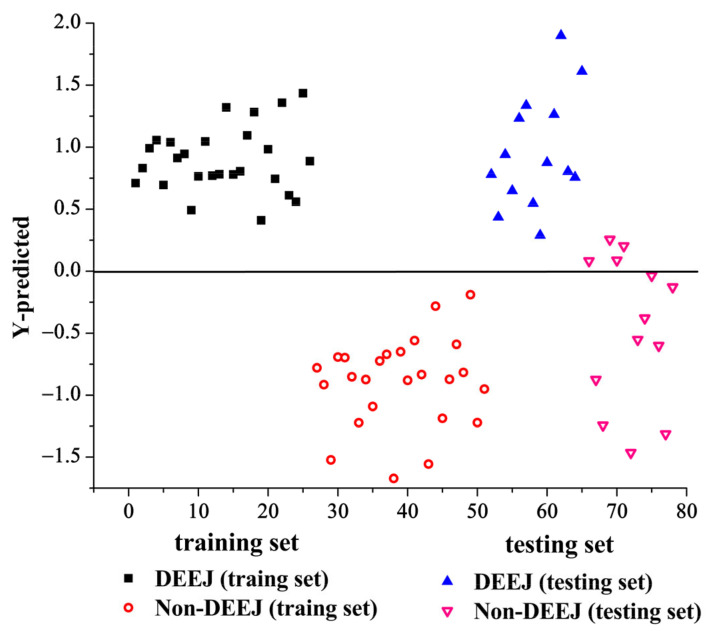
The PLS-DA model using the raw NIR spectra.

**Figure 5 molecules-28-01778-f005:**
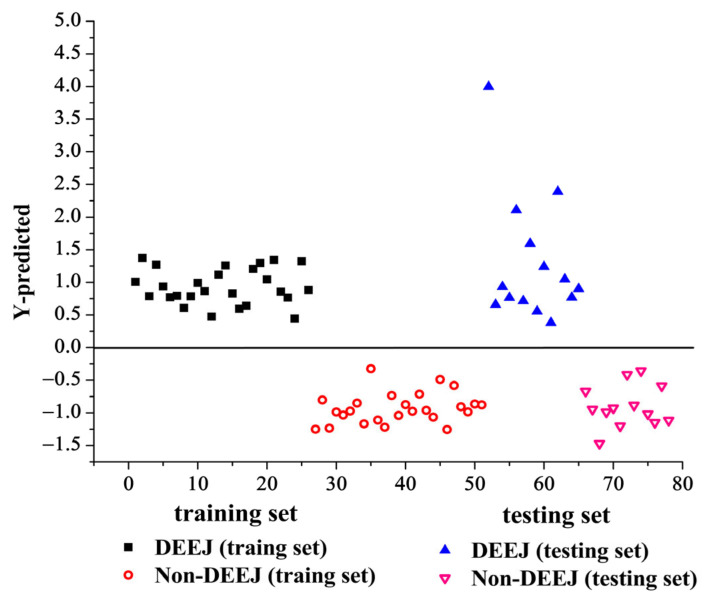
The PLS-DA model of NIR spectra pretreated by multiplicative scatter correction method.

**Figure 6 molecules-28-01778-f006:**
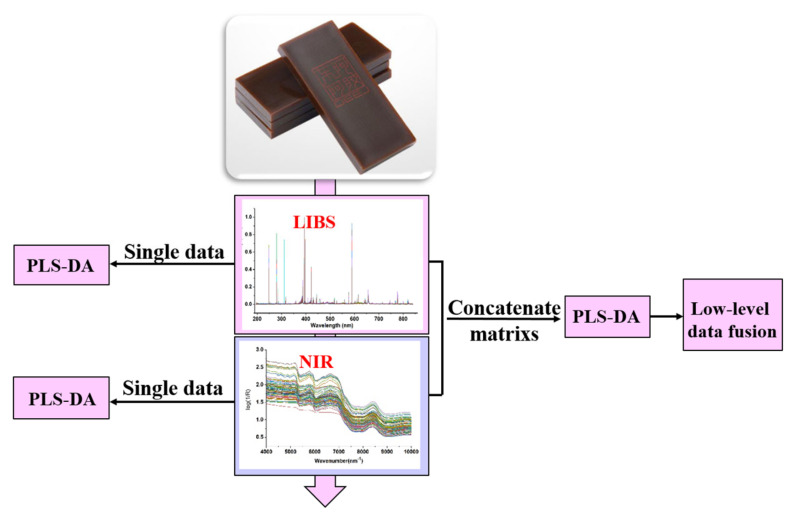
The scheme of data analysis.

**Table 1 molecules-28-01778-t001:** Selected spectral lines of LIBS Spectra.

Elements	Wavelength (nm)	Elements	Wavelength (nm)
C	247.722	Ba	455.360, 493.385
Mg	279.422, 280.124, 285.085, 383.825 516.730, 517.270, 518.359	Fe	526.987
Si	288.031	Na	588.958, 589.551
Ca	315.866, 317.921, 370.621, 393.378 396.814, 422.639, 428.287, 430.226 442.643, 443.500, 445.444, 558.850	N	742.400, 744.294, 746.927
	612.923, 616.233, 643.966, 646.212 649.394, 714.856, 720.259	Li	670.746
C-N	385.461, 386.105, 387.087, 388.302	H	656.309
Al	394.422, 396.096	K	766.515, 769.947
Sr	407.786, 421.500	O	777.216, 777.505

**Table 2 molecules-28-01778-t002:** Detailed information of all the CCA samples.

Sample No.	Manufacturer
1–40	Shandong Dong’e Ejiao Co., Ltd. Liaocheng city, Shandong, China
41–48	Shandong Huaxin Pharmaceutical Group Co., Ltd. Heze city, Shandong, China
49–61	Shandong Yanggu Guajing Ejiao factory Liaocheng city, Shandong, China
62	Shandong Dong’a Xiuyuan Ejiao biological group Neihuang Ejiao Pharmaceutical Co., Ltd. Anyang city, Henan, China
63	Shandong Jishui Ejiao Co., Ltd. Heze city, Shandong, China
64–65	Shandong Hongjitang Pharmaceutical Group Co., Ltd. Jinan city, Shandong, China
66–72	Shandong Dong’e Guojiaotang Ejiao Pharmaceutical Co., Ltd. Liaocheng city, Shandong, China
73–75	Shandong Fupai Ejiao Co., Ltd. Jinan city, Shandong, China
76–78	Shandong Yixiaotang Ejiao group Bainian Pharmaceutical Co., Ltd. Zaozhuang city, Shandong, China

**Table 3 molecules-28-01778-t003:** The discrimination results of the PLS-DA model based on NIR spectra (in the region of 5500–9000 cm^−1^) by different preprocessing methods.

Pretreatment	LVs	Training Set			Testing Set		
		Se (%)	Sp (%)	Ta (%)	Se (%)	Sp (%)	Ta (%)
Raw	7	100	100	100	93.33	100	96.3
MSC	10	100	100	100	100	100	100
SNV	7	100	100	100	100	92.86	96.3
SG9	9	100	100	100	82.35	100	88.89
1st d	6	100	100	100	92.86	92.31	96.3

Some preprocessing methods for the NIR data were compared, such as multiplicative scatter correction (MSC), standard normal variate transformation (SNV), Savitzky–Golay smoothing with 9 points (SG9), and first derivative (1st d).

**Table 4 molecules-28-01778-t004:** The discrimination results of the partial least squares-discriminant analysis models based on low-level data fusion.

Pretreatment	LVs	Training Set			Testing Set		
		Se (%)	Sp (%)	Ta (%)	Se (%)	Sp (%)	Ta (%)
Raw	7	96.15	96.15	98.04	93.33	100	96.3
MSC	11	100	100	100	100	100	100
SNV	10	100	100	100	100	100	100
SG9	13	100	100	100	82.35	100	88.89
1st d	7	100	100	100	93.33	100	96.3

Some preprocessing methods for the NIR data were compared, such as multiplicative scatter correction (MSC), standard normal variate transformation (SNV), Savitzky-Golay smoothing with 9 points (SG9), and first derivatives (1st d).

## Data Availability

Not applicable.

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
