# Peer review of "A Synergetic Strategy for Brand Characterization of Colla Corii Asini (Ejiao) by LIBS and NIR Combined with Partial Least Squares Discriminant Analysis"

_molecules, 2023, doi:10.3390/molecules28041778_

Round 1
Reviewer 1 Report
Review report
As stated in the title, this study uses LIBS and NIR spectroscopy in combination with a chemometric method, partial least squares discriminant analysis (PLS-DA) for the characterization of Colla Corii Asini (Ejiao) from different manufacturers in China. There is no doubt that the used spectroscopic methods are versatile and data-rich, and chemometric methods are of great assistance in handling such large amounts of data and gaining meaningful insights from them.
Despite the title being intriguing and promising, the content is not sufficient to support it adequately. Generally speaking, a classic methodology to analyze a sample by LIBS and NIR, and to process spectral data with chemometrics was used without any suggestions to improve what is already known in the literature. I advise the authors to add explanations and analyses to improve the paper and specially to understand the brand characterization by used method. Moreover, there is a lack of arguments in the interpretations. What is the technical advancement, the innovation, or the challenge of this study? Also, the quality of the English in this article needs to be improved.
Listed below are some comments, questions, and suggestions that need to be addressed.
Abstract
- In the abstract, we don't see what is the point to analyze this sample. (The only mention is « to discriminate the bands of E-jiao (Colla Corii Asini, CCA), a precious TCM »). The authors should add some explanations in the abstract about the purpose of this paper. The importance of work is missing in the abstract.
- Line 15, « rapid identification of traditional…. »
- Line 17, bands – brands
- Line 18, Fourty (DEEJ) and thirty-nine (non-DEEJ) samples (total of seventy-nine) were investigated for the study. But table 1 reports only 78 samples, also section 2.4 claims 78 samples. Please correct the number of used samples throughout the paper.
- Lines 23-26, The related sentences are misleading and need to be clarified. If the classification accuracy from the individual techniques is good enough, why has data fusion been performed?
1. Introduction
- Introduction of the paper is general in nature. In order for it to be effective, it must include some specific information.
- Is the primary focus of the work based on quality control? If that is the case, is it a line production? Hence the necessity of using a fast instrument for online monitoring. Please discuss.
- Line 35, « …as precious a traditional…» should be « …as a precious traditional…»
- In line 38, the use of Nowadays and were in a sentence is contradictory.
- The readers might find it helpful to have a brief explanation of the fundamental workings of LIBS and NIR spectroscopic methods, along with some key references.
- The PLS-DA method is not discussed in the introduction. Kindly include it with a few essential references, If the intended reader is a spectrophysicist or spectrochemist, not a data analyst.
- Reference 18 on line 64 is unrelated to the discussed context.
2.1 Sample preparation
- Please correct the number of used samples throughout the paper.
- Line 86, Does sample storage influence the results? It might be interesting to get an idea.
2.2 Sample measurement
- Did you perform a parametric study to optimize the experimental parameters such as delay time, gate width., laser energy, etc. in order to obtain the maximum signal-to-noise ratio?
- Line 95 and 97, what is the correct focal length used, is it 80 mm or 50 mm?
- Line 100, laser energy per pulse of 340 mJ seems to be relatively more than the required for the generation of laser-induced plasma. Can you estimate the value of laser fluence?
- Line 102, did you carry out a parametric study to estimate the appropriate number of shots and number of sites?
- A simple schematic illustration of the two spectroscopic approaches might be useful to include.
- Was the software LTB-Sophi used for acquiring the data or to analyze acquired data?
2.3 Spectral pretreatment
- Line 114, what does normalization mean here? is it normalization to the maximum intensity of some specific spectral line, or something else? Why this normalization is an important pretreatment method? Please argue and add some references.
2.4 Multivariate analysis and Latent Variable Analysis
- Line 129, « Following PLS-LDA, » does that mean, after employing PLS-DA, again several pretreatments have been performed, if yes what are these methods?
- Line 129, PLS-LDA acronym was not defined earlier, please correct.
- Line 130, what are latent variables, and how these are related to the classification and its accuracy?
2.5 Data fusion strategy
- Line 146, CAA should be CCA.
- Line 146, what is the meaning of low-level fusion strategy here? How was it achieved?
- As it is claimed that data fusion is improving the results, this section needs to be written in detail.
3.1 Spectral features of LIBS and NIR
- Is the source of the H and O lines in figure 1 the sample or the ambient air?
- It is difficult to visualize the features of spectral lines in figure 1. It would be better to subdivide spectra into multiple wavelength ranges for better visualization. Also, to get rid of non-labeled parts of spectra (e.g., 200-240 nm, 660 to 760 nm, etc.).
- Line 157, please include NIST database reference.
- Ba, Fe, Si, Al, N, and Li spectral lines have not been marked in figure 1, while reported in table 2.
- In line 172, the sentence is misleading, as it claims that LIBS and NIR spectra are depicting similar characteristics, while LIBS spectra are not showing the presence of molecular structure of water, amino acids, etc., which is shown in figure 2.
3.2 PLS-DA model from LIBS data
- You should add also the loading plot from the PLS-DA model, in order to see which spectral lines are discriminant. Also, it would be great to include a 95% confidence ellipse for easy visualization of figures 3, 4, and 5.
- It seems that there are discrepancies in figures 3, 4, and 5. According to my understanding, there should be 26 samples for non-DEEJ training and 13 samples for DEEJ testing. But it can clearly be seen that in figure 3 these numbers are 25 and 14, in figure 4, these are 25 and 13, while in figure 5 these are 25 and 11. Please explain.
- This section needs more discussion about the algorithm used and possible explanation for such classification.
- To overcome inherent biases of investigator’s mind, such analysis must be done in a double-blind manner. Were these classification analyses done in such a way?
3.3 PLS-DA model from NIR data
- Line 192-194, is it true that the accuracy obtained from NIR data, is the exactly same as that obtained in LIBS data (lines 185-187)? It seems the text has been repeated in these lines. Please check.
- In line 200, the sentence needs to be corrected grammatically.
- As the MSC method has shown the best results for NIR data, why was it not attempted for LIBS data as well?
- Lines 220 and 242, ‘*’ is not cited anywhere.
- In figure 5, what does the red circle represent?
- What is the point of combining or fusion of NIR data with LIBS data if NIR data have already successfully classified the sample to 100%?
3.4 PLS-DA model from data fuion of LIBS and NIR
- Please provide a flowchart or simple explanation of the data fusion process.
- The tables' numbers are incorrect; next to table 3, there is table 5.
- Line 236, « ..sets are all.. » 1 or 100?
- Line 238, In table 5, there is no data related to LVs=11. The best number of LVs is 10 or 11? Total accuracy reached 1.0, is it correct?
- Is it possible to provide a classification figure (similar to figures 3, 4, and 5) with fusion data? And to estimate, which part of the data has a major contribution to the classification accuracy.
Author Response
Thanks for your consideration of Manuscript ID: molecules-2085671. The comments and suggestions are very helpful and valuable for revising and improving the manuscript. We are very appreciated with the helpful comments. The revised manuscript and the responses according to the reviewers' suggestions were submitted.

Reviewer 2 Report
The work under consideration is devoted to the joint use of Laser induced breakdown spectroscopy (LIBS) and near infrared spectroscopy (NIR) for investigation of different samples of E-jiao. E-jiao is made from donkey-hide and widely used in traditional Chinese medicine. The combined application of LIBS, a method used for elemental analysis, and NIR, which provides information on molecular bonds vibration, for the characterization of products of natural origin is not a new approach (see, for example, Ref. 25 of this work). The authors of the presented article applied this method to distinguish between E-jiao samples produced by different manufacturers. The conclusion was made that “synergetic strategy of LIBS and NIR spectra demonstrated the possibility of brand discrimination of E-jiao”. But it is absolutely impossible to understand the degree of validity of this conclusion. In lines 53-54 it is said that “There were no distinct difference among the normalized LIBS spectra of all DEEJ and Non-DEEJ samples”. This statement is surprising, since the product in question is not an individual substance, but a complex mixture of substances of natural origin. Also, if the spectra of all the samples are indistinguishable, how can they be used to distinguish the samples? In lines 115-117 authors write “for NIR data, basic spectral pretreatment methods were applied…”, but in Figure 2 one can see only raw NIR spectra with ill-defined extremes. The procedures leading to obtaining the data presented in figures 3, 4, 5 should be described in detail. The reason why all non-DEEJ samples, which were produced by eight different manufacturers, are indistinguishable from one another should also be explained. It is necessary to note the extremely careless attitude of the authors to the design of the article. The text is made up of fragments with different fonts, includes many obvious errors (see, for example, lines 45, 61, 74, etc.) In my opinion, the article should be completely reworked.
Author Response

(The authors gave the same response as above.)

Reviewer 3 Report
The manuscript by Xia et al. “A synergetic strategy for brand characterization of Colla corii asini (Ejiao) by LIBS and NIR combined with partial least squares discriminant analysis” reports a study that combines LIBS and NIR spectroscopy (individually, and both via data fusion) with PLS-LDA to directly and rapidly identify and discriminate four brands of traditional Chinese medicine (TCM) on the basis of the elemental (inorganic) and molecular (organic) content. The claimed novelty is that although many spectroscopic techniques have been employed in that respect, no earlier attempt had been made to employ complementary information about the organic and inorganic composition simultaneously. However there are numerous papers in the literature combining these two techniques with chemometrics analysis and fusion in other related applications. It is my opinion that, that in itself even after the authors’ further claim that “brand characterization of CCA by combining NIR with LIBS has not yet been reported yet” constitutes enough novelty. Neither is “Also, LIBS is first used to study CCA”. The authors need to re-state what is really novel in the work.
Nevertheless the manuscript should be considered for publication once that is addressed together with the following observations.
The authors need to polish the English and grammar. See everywhere in the text where I have marked in yellow – they are self-explanatory.
E.g.:
Abstract
Spectroscopy (Spectroscopic) techniques are appealing tools for rapid identification (of) traditional Chinese medicine (TCM) with no or little sample preparation (preparation).
Fourty Ejiao samples of Dong’e Ejiao (DEEJ) samples and thirty-nine samples of other different bands were detected by LIBS and NIR (spectroscopy?). What do the authors men by brands were detected by LIBS and NIR?
· The results demonstrated that two individual techniques had good classification performance, especially the NIR.
· Compared to single data block, data fusion models also showed excellent predictive power with no misclassification.
The authors should give the quantitative performance metrics for total accuracy, sensitivity and specificity
Introduction
Line 65. “Near-infrared region mainly corresponds to vibration information of hydrogen bonds in a molecular structure, such as carbon hydrogen (C-H), oxygen hydrogen (O-H), nitrogen hydrogen (N-H) and even physical properties [20, 21]”. What physical properties are the authors referring to?
Experimental
· “The laser beam is focused on the sample surface by a microscope objective lens system with an 80 mm focal length”.
· “The focused spot size on the target surface was about 100 μm in diameter by a lens (Avantes UV-74) with a 50 mm focal length”.
Which is which?
Spectral pretreatment
“Thus, only normalized preprocessing was performed on LIBS data to reduce signal uncertainty”. The authors should elaborate how this was done.
Line 200. What do the authors mean by “Due to not all wavebands constructed the NIR spectra of CAA samples possess the special discrimination ability”?
Where is Table 4?
General comment
What advantage does data fusion provide to the study compared to the results obtained individually by NIR and LIBS?
Author Response

(The authors gave the same response as above.)

Round 2
Reviewer 1 Report
I appreciate the efforts of the authors in improving the quality of the article after revision. Unfortunately, I cannot find the author's response to the first review report and questions. However, some of the questions from the first revision are still unattempted/unaddressed.
Listed below are some comments, questions, and suggestions that need to be addressed.
1. Introduction
- The readers might find it helpful to have a brief explanation of the fundamental working principles of LIBS and NIR spectroscopic methods, which is missing in the current version of the manuscript.
2.1 Sample preparation
- Line 90, Does sample storage influence the results? It might be interesting to get an idea.
2.2 Sample measurement
- Did you perform a parametric study to optimize the experimental parameters such as delay time, gate width., laser energy, etc. in order to obtain the maximum signal-to-noise ratio?
- Line 102, did you carry out a parametric study to estimate the appropriate number of shots and the number of sites?
- A simple schematic illustration of the two spectroscopic approaches might be useful to include.
- Was the software LTB-Sophi used for acquiring the data or to analyze acquired data?
- Line 109, analysised should be "analyzed".
2.3 Spectral pretreatment
- Line 117, what is the meaning of poor repeatability here? Could you please provide an estimation of repeatability in terms of standard deviation?
2.4 Multivariate analysis and Latent Variable Analysis
- Line 134, Is it PLS-LDA or PLS-DA?
3.1 Spectral features of LIBS and NIR
- Is the source of the H and O lines in figure 1 the sample or the ambient air?
- Some of the spectral lines (e.g., Si, Li, Ba, C-N,..) are not labeled in figure 1, while reported in table 2. Also, the quality of figure 1 is not high.
3.2 PLS-DA model from LIBS data
- You should add also the loading plot from the PLS-DA model, in order to see which spectral lines are discriminant. Also, it would be great to include a 95% confidence ellipse for easy visualization of figures 3, 4, and 5.
- It seems that there are discrepancies in figures 3, 4, and 5. According to my understanding, there should be 26 samples for non-DEEJ training and 13 samples for DEEJ testing. But it can clearly be seen that in the figures, these numbers are 25 and 14, Please explain.
- Figure 3, what could be the possible explanation for the data point which is completely out from (the non-DEEJ testing set, top right corner of the figure)?
- This section needs more discussion about the algorithm used and possible explanation for such classification.
- To overcome inherent biases in the investigator’s mind, such analysis must be done in a double-blind manner. Were these classification analyses done in such a way?
3.3 PLS-DA model from NIR data
- As the MSC method has shown the best results for NIR data, why was it not attempted for LIBS data as well?
- What is the point of combining or fusion of NIR data with LIBS data if NIR data have already successfully classified the sample to 100%?
3.4 PLS-DA model from data fuion of LIBS and NIR
- Please provide a flowchart or simple explanation of the data fusion process.
- Line 236, "When the latent variables are 11, all the Se, Sp, and total accuracy reached 100%", Why the data related to LV =11 was not reported in table 4?
- Is it possible to provide a classification figure (similar to figures 3, 4, and 5) with fusion data? And to estimate, which part of the data has a major contribution to the classification accuracy.
Conclusions
- Line 244-245 is not grammatically correct, please rewrite.
- Please specify the key obtaining of this article in a concrete way.
Author Response
Thank you for your help. We appreciate the positive comments about the manuscript. The Manuscript was reviesed according the suggestions in a concrete way.

Reviewer 2 Report
The manuscript was improved and can be accepted for publication after the English edition.
Author Response
Thank you for your guidance and help.
Round 3
Reviewer 1 Report
I commend the authors for their efforts in improving the manuscript, and I agree with its publication in the journal upon addressing the following minor suggestions:
- In section 3.2, Line 204, it says sample number 38 belongs to DEEJ, but in figure 4, it is Non-DEEJ red circles. Please clarify or correct.
- Also, just looking at spectra, it is difficult to judge if samples 38 and 74 are similar. May be some statistics can help in justifying this.
- If these two are similar why (in figure 4) the y-predicted value of sample 38 is negative around -1 and for sample 74 it is highly positive around 1.5? Please clarify.
Conclusion
Line 271 ("We also ...LIBS data".) is not a part of the conclusion it should be in the data handling or discussion section.
Author Response
Thanks for your consideration of Manuscript ID: molecules-2085671.
